# Urbanization and Vulnerability of Architectural Heritage: The Case of Dar es Salaam CBD

**Swai Ombeni \*, Dorothea Mbosha** 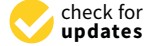 **and Simon Mpyanga**

Department of Architecture, School of Architecture, Construction Economics and Management, Ardhi University, Dar es Salaam P.O. Box 35176, Tanzania; dorothealaswai@gmail.com (D.M.); smpyanga@gmail.com (S.M.)
\* Correspondence: swaiarcht@gmail.com

**Abstract:** The architectural heritage present in Dar es Salaam Central Business District (CBD) spans across Arab, German, British, and post-colonial eras. The city is rich in buildings with combined architectural styles. Over the past few decades, Dar es Salaam has been experiencing considerable pressure from urbanization. This has resulted in a boom of contemporary construction approaches, yet little concern has been given to the existing old buildings and historical fabric in general. Although urbanization is an unstoppable reality due to the forces it carries with it, it is necessary to find ways to strike a balance between urbanization and its impact on the original urban setting which is less explored in Dar es Salaam. This study investigated the relationship between urbanization and architectural heritage with the intention to balance the two, and set to answer questions about how the two can co-exist. Through a case study approach, changes such as physical transformation, elimination, and replacement of architectural heritage buildings were investigated and analyzed through maps, graphs, and charts. The results have indicated that the driving forces of urbanization such as population, policies, and economy have been influencing each other in physical transformation and demolition of architectural heritage buildings throughout the period between 1967 and 2020. The study suggests that activities such as ecotourism which will enhance social economic benefits should be promoted to support both the urbanization process and architectural heritage conservation.

**Keywords:** urbanization; heritage; architectural heritage; urban form; built fabric

## 1. Introduction

Urbanization has been regarded as one of the major forces currently experienced by the whole world. By 2018, 55% of the world's population was living in urban areas [1]. This trend is expected to increase to 68% by 2050 whereby 2.5 billion people are expected to be in urban areas, most of them in Asia and Sub-Saharan Africa [1]. The built environment has become transformed in the urbanization process due to increased demand of services such as transport, clean water, energy, adequate housing, sanitation and manufactured products [2]. Due to such needs, governments and the private sector have been forced to invest in the construction industry in large scale, replacing old buildings with new ones in order to cater for infrastructure and service needs [3]. Zhang et al. [4] noted that the construction industry is the major contributor to factors associated with social, economic and environmental depletion. For this reason, much attention is being paid to issues concerning environmental depletion where focus has been on the construction industry, to reduce the impact it causes to the environment. Rodwell [5] suggests that one way of ensuring environmental sustainability is to conserve existing buildings including historical buildings and sites.

The situation in Tanzania, according to NBS Census [6–9], shows that the number of people living in urban areas is 44.9 million. With the historical trend from 1967 to 2012, there has been an increase of 12 million people in urban areas. The predicted rate of urban population growth is 1.4 million people between 2012 and 2050, which is twice the rate of total population growth. This means that in 25 years, more than half of the population

in Tanzania will be living in urban areas [3]. To react to such development of events, the construction industry has been reported to have a growth above the general economy and has responded to investments done in infrastructural, residential and commercial establishments [2]. The demand has pushed for new architectural interventions with the expectations that old buildings will sacrifice their heritage values [1].

Dar es Salaam is a city in Tanzania located along the East Coast of the Indian Ocean, and it is among the fastest growing cities experiencing rapid urbanization in Sub-African Africa [10]. The city accommodates 10% of the national population and 30.3% of the country's urban population. Its population is expected to double from 4.4 million people in 2012 to 10.8 million people in 2030 [3]. The history of Dar es Salaam dates back to the 8th Century when it was a small native settlement [11,12]. In mid-19th Century, the city started expanding, when Sultan Majid of Zanzibar initiated a plan and constructions for a new town by the large inland harbor close to a small native village of Mzizima, and then grew further through German, British colonial periods and post-colonial period [13,14]. The city is therefore rich in buildings with a combination of architectural styles. Apart from Arab, German and British architecture, the remnants of native Swahili and Indian buildings add to the asset of architectural heritage of the city, which creates a sense of legibility and demonstrates the age of the city to its inhabitants [13,14].

In Dar es Salaam, architectural heritage conservation has been managed by various organizations under both government and private. Several lists have been given out by the Department of Antiquities under the government Ministry of Natural Resources and Tourism (MNRT), Dar es Salaam Master Plan Consortium under the government Ministry of Lands, Housing and Human Settlement Development and by DARCH (Dar es Salaam Centre for Architectural Heritage) which is a private non-profit organization dealing with conservation of architectural heritage. The lists have been contradicting one another as they are not reconciled as one resulting to conflicts on managing architectural heritage buildings giving power to urbanization due to lack of a consistent list [15].

Over the past few decades, Dar es Salaam has been experiencing rampant pressure from urbanization [10,16]. This has resulted in a boom of contemporary construction approaches with little concern on the existing old buildings and historical fabric in general. This is manifested in the abandoning of once used historical buildings to a state of no repair. The intertwining of architectural styles provides a sense of place that evokes life before the present and contributes to the character that serves as an intangible asset [16,17]. The appearance of building materials has blended to the streetscapes—carved timber doors and windows, decorative ironwork, timber facades, wooden railings, roofing tiles, etc. The original building heights at the CBD balanced the skyline where common buildings remained humble, giving room to religious buildings such as temples, churches and mosques to dominate the silhouette and serve as points of reference for the city [18].

Although urbanization is an unstoppable reality due to the forces it carries with, it is necessary to find ways to strike a balance between urbanization and the impact it carries to the original urban setting which is less explored in Dar es Salaam. Researches and literature on the development of Dar es Salaam have been done; nonetheless, few describe explicitly the expenses the architectural heritage incurs to accommodate the pressure brought about by the urbanization process. Amar [2] investigated stakeholders' perceptions in Australia and Tanzania on the conservation of cultural built environment. Kigadye [10] studied architectural heritage in rapid urbanizing cities focusing on legislative and institutional frameworks for management of conservation areas in the city center of Dar es Salaam. These studies were based on decision making aspects which include a legislative framework with little regard for what happens on the ground.

Despite the important roles that urbanization and architectural heritage play and existence of the relationship between the two fields, little has been studied to establish this relationship and how the two can co-exist. Unceasingly, interventions have been implemented by different stakeholders [2]. It has also been observed that the interventions lack reconciliation [19]. This study sought to investigate the relationship between urbanization

and architectural heritage with the aim of trying to strike a balance between the two, and answer the question about how the two can co-exist. The study hoped to add knowledge about socio-economic potentials of architectural heritage as a basis of welcoming new economic paradigms brought by urbanization. The findings and conclusions made will help in making informed decisions in handling the two processes, in order to have a common goal with respect to attaining urban socio-economic improvement. It will also benefit a wide range of stakeholders, from both private and public sectors, involved in urban design, urban planning, research, tourism and city dwelling.

## 2. Materials and Methods

### 2.1. Conceptual Framework

The study employed morphological analysis on studying urbanization and architectural heritage. The approach focused on building level, street level and city level [20–22]. At building level, the information was for the purpose of: (i) studying the housing and social economic demands of architectural heritage buildings in the process of urbanization, and (ii) determining the strategies that have been used for the coexistence of urbanization and architectural heritage. At street and city levels, the study focused on information related to appearance and functionality of streets and the city in general, as influenced by development policies, population, and economy, and how the two have been merged by such influences [20–22]. Architectural heritage vulnerability acts as a dependent variable to urbanization, as illustrated in Figure 1.

### 2.2. Methods

To establish a link between urbanization and architectural heritage, a case study within Dar es Salaam CBD was conducted. Dar es Salaam CBD was chosen as it is an active area with a substantial number of architectural heritage sites where rapid changes are taking place. Again, it is neither a world heritage site nor is it on a tentative list. A criterion for choosing buildings from each ward was determined.

The city was selected because it is an area with a good number of architectural heritage buildings, which are under pressure of demolition, and has a good number of new structures as a manifestation of clear physical transformation. It is also multi-faced in terms of different types of buildings under urbanization. Data from literature review and observation was employed in the analysis of this study [23,24]. Literature review was done on public documents such as acts, policies, laws, exhibition panels, reports, journals, master plans, satellite maps and statistical data.

Buildings identified as being of architectural heritage in this research are based on the list given out by the Department of Antiquities (DOA) in 1995, with 28 buildings to be under conservation, 27 of them located in Dar es Salaam CBD. The list given out by the department of Antiquities in 2006 with a total of 110 buildings was controversial, and a new list was supposed to be given out by the task force created but no list has been given out to date, list provided by DARCH (Dar es Salaam Centre for Architectural Heritage) with 58 buildings, 52 still standing and 6 buildings demolished and list by Dar es Salaam Master Plan Consortium 2012–2032 which has a total of 259 buildings located at the CBD. All the lists are considered valid. The provision of lists has been guided by The Antiquities Act of 1964 and its amendment in 1979 [15,25]. It was realized that despite of existence of the lists a detail description of the degree and extent of conservation had never been clearly stated. Until recently in 2011, the Dar es Salaam Master plan 2012/2032 established a list of 259 buildings to be under conservation and has documented location and extent of alteration i.e., renovation, renovation for 99 buildings among them but not yet under implementation due to the issue of mandate on who has power over this.

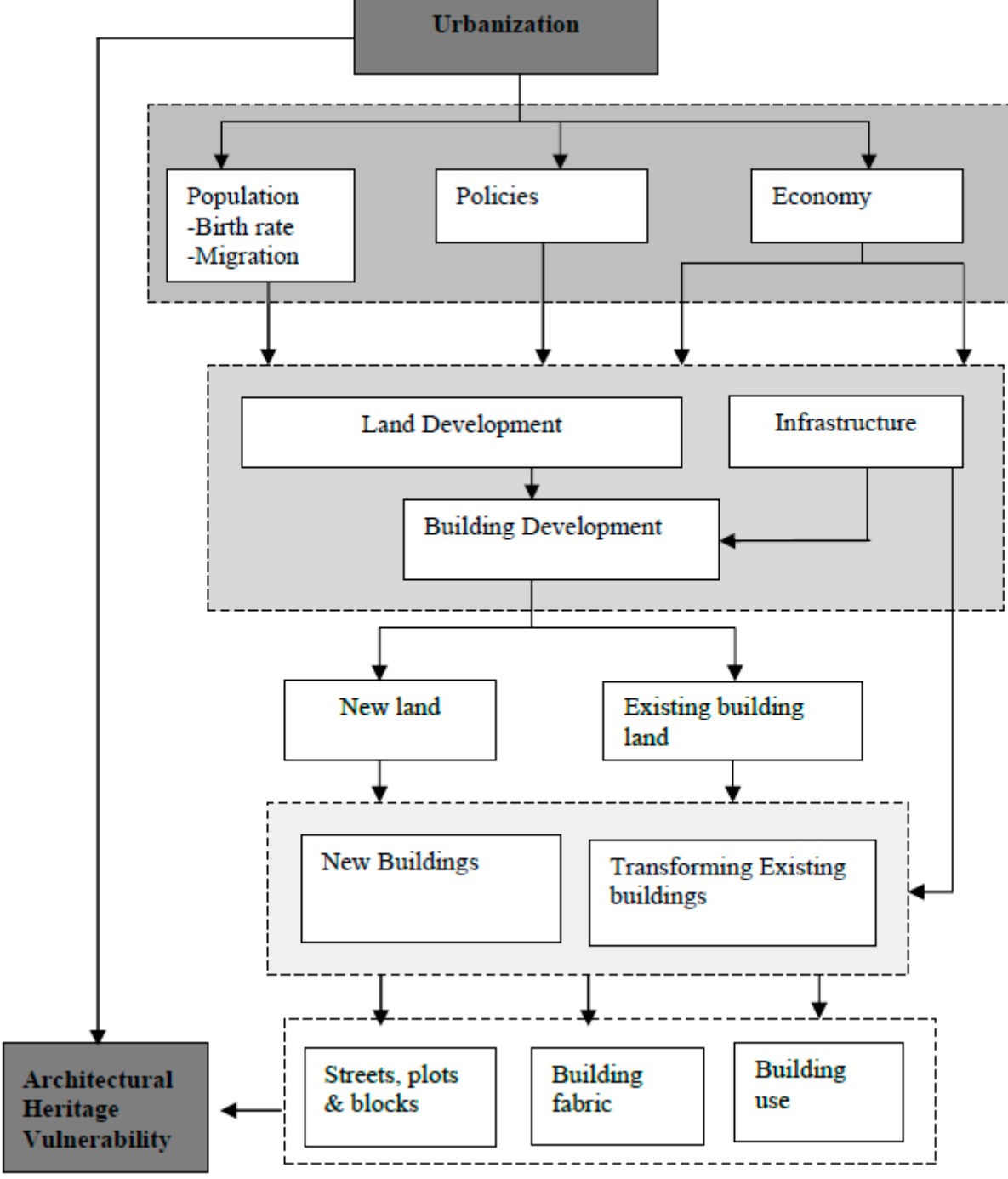

**Figure 1.** Conceptual framework (Source: Author, 2020).

Architectural heritage elements present in the case study area were observed, and a photo registry was consulted; these were related with satellite images, aerial photographs, and auto photographs. Maps were reconstructed from aerial photographs and Google earth satellite images of 1967, 1994, 2005, 2010, 2015 and 2020 to identify the demolitions and the newly-built structures with the interval between 27, 11, 5, 5, 5 years respectively. The analysis started in the 1960s because they are years when the country had just obtained its independence in 1961. It was a period of establishment of new policies such as Arusha Declaration of 1967 which had big impact to the built fabric of Dar es Salaam CBD. Threats of demolition of buildings had also started despite of the existence of Antiquities Act of 1964 to protect architectural heritage buildings. The intervals of the years are not the same due to the challenge of obtaining aerial photographs and satellite images. Google earth

satellite images of the study area of Dar es Salaam CBD are available after the year 2000, which is after the beginning of the new millennium of 21st Century. Hence, aerial images are used for the years 1967 and 1994.

## 3. Results and Discussion

### 3.1. Physical Transformation of Architectural Heritage Buildings

Physical transformation in this study includes changes in facades in terms of colors, additions and removal of structures and objects which were available when the building was built. Due to the influence of urbanization, historical buildings have been altered to fit with functions assigned to them due to new needs caused by urbanization (refer to Figures 2 and 3). The needs include residential, office and commercial space.

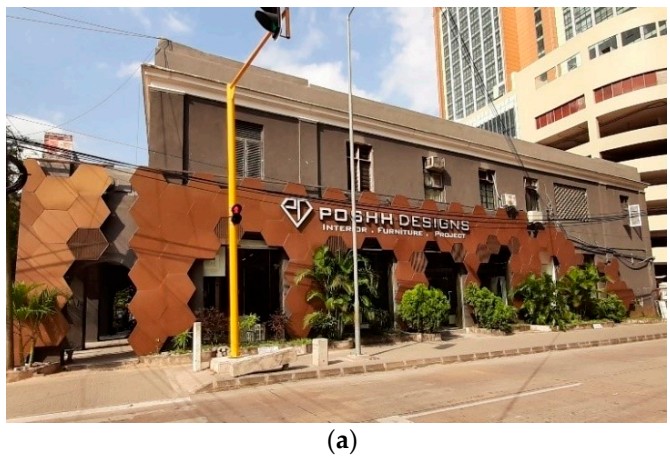 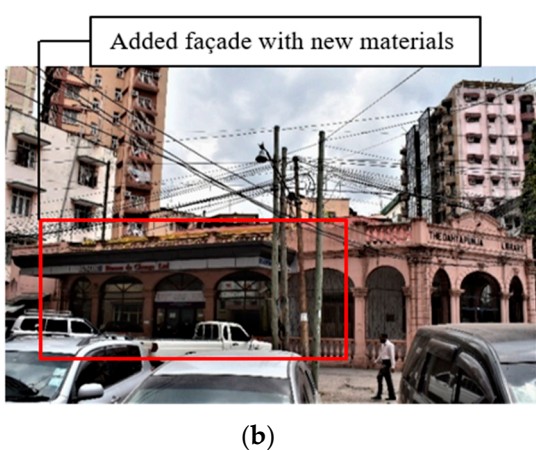

(**a**) (**b**)

**Figure 2.** (**a**) Building at the corner between Samora Avenue and Morogoro Road with facades changed to accommodate current functions. (**b**) Dahya Punja Library built in 1928 for Gujarat community at the corner of Indira Gandhi St. and India St. with parts of facades changed to modern materials. (Source: Author, 2020).

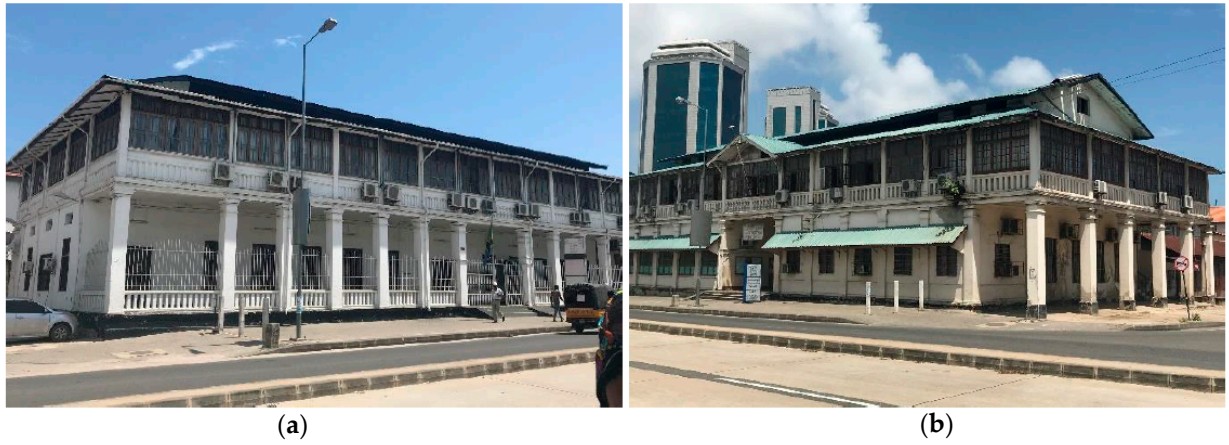

(**a**) (**b**)

**Figure 3.** (**a**) NBS building (**b**) Survey and Mapping Division building; Series of German buildings with balconies covered to increase the floor plan area. (Source: Author, 2020).

Many commercial streets have been observed to have buildings with constantly changing facades especially for the ground floors to accommodate commercial activities. This has been the case in many buildings along Morogoro Road. Buildings along Kivukoni Road (German buildings) have had their functions changed, and some parts of the buildings such as balconies have been covered to increase indoor workable space.

### 3.2. Elimination and Replacement of Architectural Heritage Buildings

Between 1967 and 1994, as noted, at least 9 buildings around Samora Avenue, Bridge and Kaluta streets were demolished, and at least 12 new buildings were built. Among the 12 built buildings, at least 9 were built on unbuilt sites and 3 were built on sites where old buildings had been demolished (Figure 4 below).

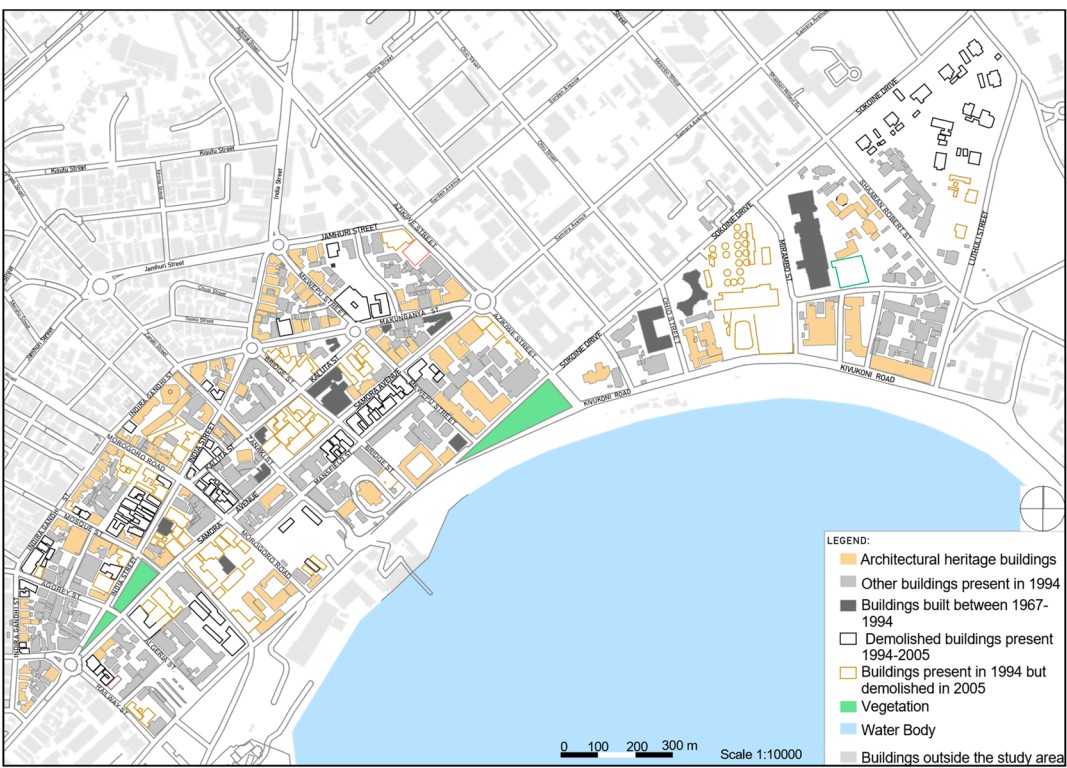

**Figure 4.** Demolitions and new buildings between 1967 and 1994 (Source: Author, 2020. Modified from a satellite image of 1967 and an aerial photograph of 1994.

Some of the buildings built between 1997 and 1994 are Sukari House at the junction between Ohio Street and Sokoine Drive, built around 1988; NIC Investment building (LIFE House) at the junction between Ohio Street and Sokoine Drive, built around 1980's; the Bank of Tanzania Building along Mirambo Street [26], built around 1970's (this was demolished in early 2000's and built anew); Ministry of Water building located at the junction between Sokoine Drive and Mkwepu Street, built around 1978; Extelecoms building along Samora Avenue and Mkwepu Street, built around 1970's; and Masdo building at the junction between Makunganya and Samora Avenues, built around 1970s.

Between 1994 and 2005 (a period of about 11 years), at least 57 buildings were demolished, and at least 22 new buildings were built on sites where old buildings had been demolished (Figure 5. Buildings built around this time were of mixed ownership; these include Hyatt Regency, Dar es Salaam (Kilimanjaro Hotel) located along Kivukoni Road. This hotel was built around 2005 after the demolition of the old building of the same hotel owned privately. The Bank of Tanzania twin towers located along Mirambo Street and Sokoine Drive were built around 2005 after the demolition of an earlier old structure. These twin towers were the first high rise glass structures in the city. The buildings are owned by the Government of Tanzania. The plot behind the State House, along Luthuli Street, was previously dominated by high class residential houses. About two of these structures were demolished during this interval and replaced by new government institution buildings. Along Indira Gandhi Street, at the site of Khoja Shia Ithnasheri Mosque, a building was added, privately owned by Indian communities, while NHC built a new tall structure at the corner between Samora Avenue and Morogoro Road.

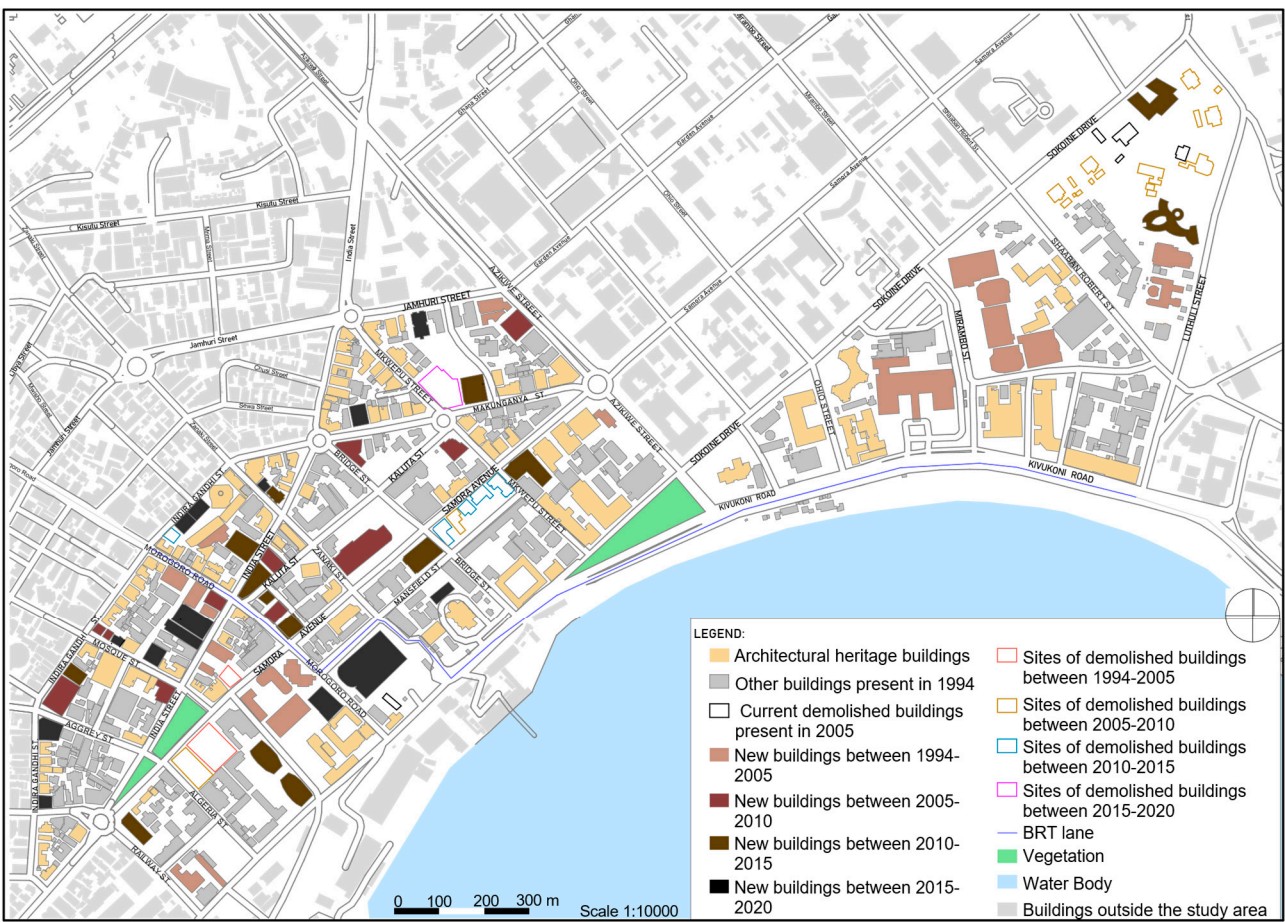

**Figure 5.** Demolitions and new buildings between 1994 and 2020. (Modified from satellite images 1967, 1994, 2005, 2010, 2015 and 2020) (Source: Author, 2020).

Between 2005 and 2010 (a period of 5 years), at least 25 buildings were demolished and at least 11 new structures built (Figure 5). During this time, the private sector was building at a faster pace than corporations and the government. Awareness about architectural conservation had risen. Many protected buildings which were demolished were identified by names such as Bagamoyo House along Morogoro Road, identified by Markes [15]; Registry building along Makunganya Street, located at the corner between Samora Avenue and Railway Street was demolished in 2008. New buildings built during this time and identified by names are NHC House located along Samora Avenue, and owned by a corporation; Diamond Plaza building at the corner of Indira Gandhi Street which is a residential apartment, and privately owned; and Rainbow Hotel, which is situated along Morogoro Road, and privately owned. Others include Elite Tower, along Azikiwe Road, which is a privately owned commercial tower; LAPF Pensions Fund at the corner between Mkwepu and Kaluta streets, a high-rise commercial building which is owned by a corporation. At this interval, many of the buildings that had been demolished and built anew were located around and beyond Samora Avenue from the Indian Ocean. Most of these are privately owned.

Between 2010 and 2015 (a period of 5 years) at least 7 buildings were demolished and at least 16 buildings built (refer to Figure 5). The private sector had increased the rate of building followed by corporations and the government. Some demolished buildings included Quality Shop, located along Samora Avenue, which was demolished in 2011; Light Corner Building also along Samora Avenue, demolished in 2011; Blaschke House along Samora Avenue, demolished in 2011; and Salamander building (a famous restaurant) located along Samora Avenue, demolished in 2013. During this time there was a major

construction project underway—The Bus Rapid Transport (BRT) project. New bus lanes were constructed along Morogoro and Kivukoni roads in the study area. This project changed the use and facades of buildings and streets where it was being carried out. The street turned more commercial and very many buildings had their ground floors used for commercial purposes. Glass and Aluco bond were used as façade materials. Some buildings which were constructed around this time were PSPF twin towers along Mission Street, Samora Tower along Samora Avenue and Rita Tower located along Makunganya Street and Simu Street. This building replaced an old building which had been demolished. Government institution buildings were also built; these included the Tanzania Commission for AIDS building, along Luthuli Street, and four commercial/residential buildings along India Street. Blaschke Building was a replacement by a commercial tower which currently houses Exim Bank; besides, the Golden Plaza building along Indira Gandhi Street was replaced by Solitaire Plaza, a commercial residential tower. At the corner between Samora Avenue and Bridge Street the previous Light Corner building has been replaced by a commercial residential tower owned by NHC, also known as Samora Tower.

Between 2015 and 2020, the rate of demolition decreased compared to the past five years. At least 3 buildings were demolished, and 14 new buildings were built to replace previously demolished buildings (Figure 5). Some of the demolished buildings were Billicanas building located at the corner between Mkwepu Street and Makunganya Street. The site, which remains unbuilt, is used as a temporary car parking space. Two other buildings were demolished along Samora Avenue. New buildings include Rotana Hotel, located along Morogoro Road and Mansfield Street; DCL Commercial Bank building, located along Morogoro Road; Golden Tulip Hotel, located along Jamhuri Street; and a high-rise accommodation building still under construction within the premises of St. Joseph Cathedral. Most of these buildings are under private ownership.

Between 1967 and 1994 (refer to Table 1, Figures 6 and 7, more buildings were built than demolished. This means that there was still unbuilt land around the study area. Between 1994 and 2005, the number of demolitions surpassed the number of built structures as already explained; which shows that due to shortage of land, old buildings had to be demolished to give way to new buildings, although the number of newly built structures did not fill all the empty spaces that had been created. Between 2005 and 2020, the number of new buildings surpassed the number of demolitions as shown in Table 1, Figures 6 and 7. The curve indicates that the number of new buildings rose continuously until 2020, but such buildings have not been able to fill the spaces previously occupied by demolished buildings. This means that after all empty spaces of previous demolitions have been built, demolitions will start again to create empty space for new buildings.

**Table 1.** Summary of demolitions and new buildings (1967–2020).

| Years | Interval | No. of Demolitions | No. of Built Structures |
|---|---|---|---|
| **1967–1994** | 27 | 9 | 12 |
| **1994–2005** | 11 | 57 | 22 |
| **2005–2020** | 15 | 35 | 41 |

(Source: Author, 2020).

### 3.3. Policies, Population and Economy

Based on mapping and statistical results, it has been noted that themes have been repeating themselves. The study variables, i.e., population, policies and economy, have been influencing each other. It has been observed that between 1967 and 1994, that displacement of architectural heritage buildings was not directly impacted by population increase or economy but rather by policies. Since Tanzania was under a socialist leadership from 1961 to 1985, the guiding policy of socialism and self-reliance led to the formulation of the Building Act 1972, which nationalized private properties in the study area. This discouraged private investment in housing or factories as many private individuals stopped building.

The speed of urbanization decreased following apprehension to develop the real estate, because of fear of losing properties as it had happened in 1972. The policy also led to a decrease of population in Dar es Salaam CBD since people were required to live in socialist villages. Displacement of buildings and construction of new ones was not left to the private sector but rather to corporations, which were formed under the same policy of real estate development. Most new buildings in the study area were built by corporations such as NHC, NIC, postal authority, corporate banks, and the government itself.

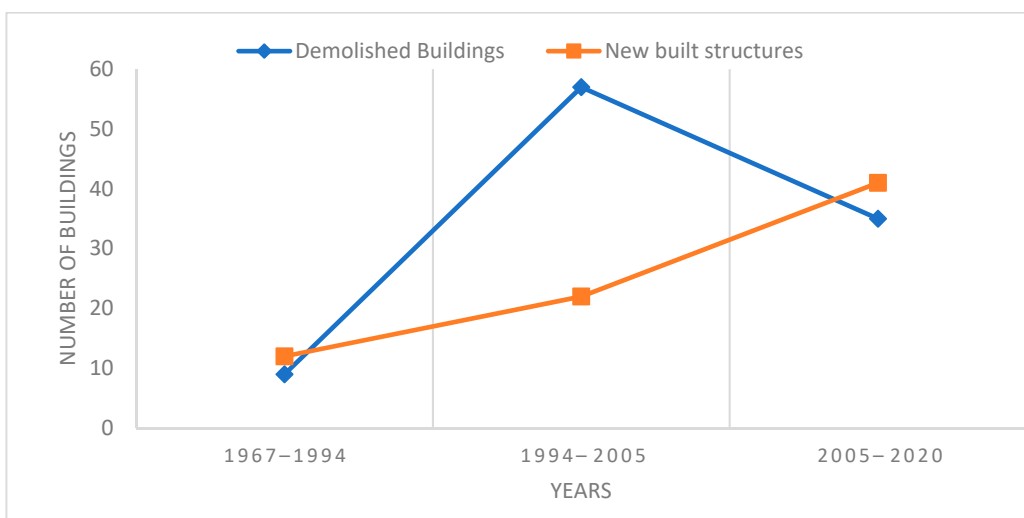

**Figure 6.** Summary of demolitions and new buildings (1967–2020). (Source: Author, 2020).

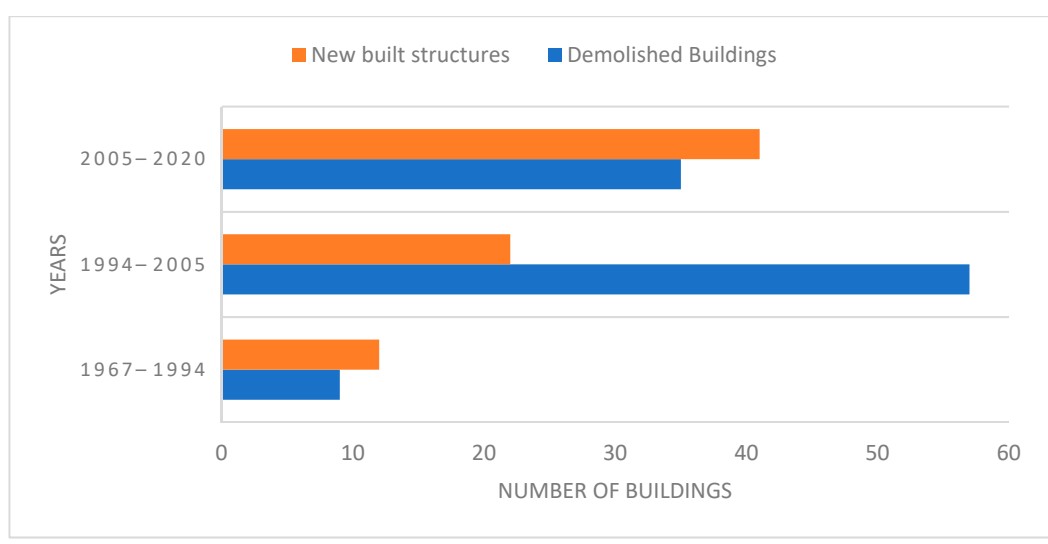

**Figure 7.** Summary of demolitions and new buildings (1967–2020). (Source: Author, 2020).

It has been noted that in the period between 1994 and 2005, displacement of buildings was influenced by policies and urban economy which determined the type and size of the urban population. Due to shift from a command economy to a free market economy by trade liberalization and privatization, Ushirika (Corporations) and the private sector acquired buildings and plots in the city center and the fear of building had decreased hence new materials, technology, design styles and designers were imported, unlimited, to the country. Some of the properties which had been nationalized were reclaimed by their owners. Corporations such as NHC increased its number of properties. This time, builders were the government itself, corporations, and the private sector. Displacement of rich individuals was so radical because the policy favored them. Hence, policies and the urban economy seemed to control the changes that took place on architectural heritage.

Between 2005 and 2020, central area redevelopment plan was the major influence for changes. It lacked protective measures of architectural heritage as its focus was to cater for the then existing and future demands for infrastructure services, daytime population; the streets had become narrower and could not accommodate the population, increasing demands for commercial and office spaces to have corporate buildings of many storeys as possible. The policies of privatization and liberation had been translated into a master plan. The guiding policy for development of Dar es Salaam CBD was the Central Area Redevelopment Plan 2000, which was used by planners and developers [27]. Since its implementation in 2002, it has completely changed the built form to an uneven skyline which seems to contradict development conditions. The current skyline of Dar es Salaam CBD is a result of the Central Area Redevelopment Plan, 2000.

## 4. Conclusions and Policy Implication

### 4.1. Conclusions

This study was derived from the curiosity of the expense that architectural heritage incurs to accommodate pressure usually exerted by urbanization. It sought to explore the trend of urbanization and its corresponding implications on architectural heritage and explore how the two can co-exist. The findings have demonstrated that urbanization is fuelled by factors such as population, policies, and economy. In the hypothesis they were noted as separate variables, but through the investigation conducted on this study they have been shown to be interlinked as they influence each. For instance, policies have been seen to influence population increase, as Ujamaa (socialism) policy led to a decrease in the number of people in the CBD, which as a result, few changes were noticed as far as architectural heritage was concerned. Population in the city has also affected policies on building uses such as allowing high rise buildings to accommodate high populations after demolition of architectural heritage buildings. Hence, the results of their influences have been documented together.

This study has also proven the importance of contextual studies as noted by Moudon [20] on the dynamic nature of cities as driven by varying social-economic forces such as urbanization. In this way, it becomes easier to address the challenges that may arise using contextualized rather than generic approaches. Though international organizations such as UNESCO and ICOMOS have come up with recommendations for conservation of architectural heritage, if these are not contextualized, they may not be applicable in some contexts.

### 4.2. Policy Implication

The Historic Urban Landscape [28] proposes that, to balance between architectural heritage conservation and urbanization, one could use the landscape approach with the goal to ensure sustainable development in the city on a wider context. Providing new functions such as service-oriented activities for example ecotourism in this context can provide social and economic benefits and provide accommodation in cities. This has been seen from the trend between 1995 and 2020 whereby the changes in building fabric was influenced by its ability to generate income, hence balancing with the economy. Further, planning of the city should be extended from being monocentric to being polycentric. Harmonious activities to ensure livelihood should be left to the old part of the city, and other activities should be kept in the new city. For example, in India, there is Old Delhi and New Delhi; in Zanzibar, there is Stone Town and New town, both of which coexist with harmonious functions. The growth of the city can be seen through this manner. Activities like wholesale business should be carried out at the CBD and create other markets at the periphery of the city and leave the CBD for wholesale (if necessary) because demand is what calls for new buildings to be built on sites that harbored old buildings. Regarding important decisions and policies concerning architectural heritage buildings and streets, a bottom-up strategy is suggested in understanding users and their needs. This would enable the design of policies which are friendly and owned by users themselves. Also, participatory approach policies involving owners and government to be formulated to protect few remaining buildings. The study

revealed lack of coordination between different stakeholders involved in conservation of architectural heritage and urban planning, which has led to generation of contradicting policies on both sides. Hence the study suggests cooperation from both sides in decisions and plans concerning land and building uses. Further, it was revealed in the study about the lack of a clear list of buildings under conservation which has led to contradiction on conservation issues. The responsible ministries, authorities and stakeholders are advised to coordinate together and provide a clear list of buildings to avoid contradictions.

**Author Contributions:** Conceptualization, S.O.; Data Collection D.M.; Proofread funding, S.O.; Reading S.M.). All authors have read and agreed to the published version of the manuscript.

**Funding:** This research received no external funding.

**Institutional Review Board Statement:** The study was conducted according to the guidelines of the Ardhi University research and publication policy and the ethical review and approval was waived in this study because it mainly involve buildings, conservation and urbanization.

**Informed Consent Statement:** All authors give consent for publication of the Manuscript.

**Data Availability Statement:** The study did not report any data, however, in case of a need, data can be available by contacting the corresponding author.

**Conflicts of Interest:** The authors declare no conflict of interest.

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
