# Peer review of "Urbanization and Vulnerability of Architectural Heritage: The Case of Dar es Salaam CBD"

_conservation, doi:10.3390/conservation1030017_

Round 1

Reviewer 1 Report

The article proposes in its introduction an important problem of loss of heritage values ​​in the case of study cited and makes a detailed description of the process of transformations suffered in recent decades. The causes and impact of the new developments linked to policies, population and economy are cited and well defined, however, we consider that the following are missing:

- More detailed description of the heritage values ​​of the existing architectural heritage through objective values: regulated degrees of protection, cataloging or inventories of typological or historical analysis. Or in the absence of such studies, analyze the problem of a lack of inventory and protection.

- In the map 1, there are several buildings named “Architectural Heritage Buildings”. Who named this buildings like this? Based on what?

- Analyze as a cause, the probable lack of heritage protection measures in the master plan "Plan for remodeling the central area 2000", and not only as a consequence.

With this approach, it will be possible to better assess the value of the architectural heritage of the cited case study, and what are the specific problems that have not been preserved correctly. In short, it is considered necessary to provide a more detailed heritage evaluation to really understand this duality between urbanization and architectural heritage, to be able to understand this proposal for future coexistence.

As minor comments, it is recommended:

- Explain what the acronym CBD means, since it is not explained anywhere, and any international reader may not understand what it refers to. It could be explained in the introduction and in the rest of the paper the acronyms could be used.

- Explain why the date of 1967 is used as the beginning of the analysis (what milestone occurs on that date?), Since in the first periods, the transformation process does not materialize so strongly.

- Be careful with proposing tourism as an exemplary measure to revitalize and preserve heritage, since mass tourism is being precisely one of the causes of the destruction of heritage in other parts of the world. It is recommended to specify sustainable and controlled tourism measures.

- The English language could be improved

In short, it is considered an interesting case study, and the dissemination of the research results may be useful for society and the scientific community, however, we consider it necessary to complement the urban and socioeconomic analysis with a more detailed heritage approach.

Author Response

Response to Reviewer 1 Comments

POINT 1

The name of Author #2 is not DOROTHEA LASWAI

Response 1:

Author #2’s name is DOROTHEA MBOSHA

POINT 2

A more detailed description of the heritage values ​​of the existing architectural heritage through objective values: regulated degrees of protection, cataloguing or inventories of typological or historical analysis. Or in the absence of such studies, analyse the problem of a lack of inventory and protection.

Response 2

In Tanzania (Dar es Salaam) architectural heritage conservation has been managed by various organs both under the government and private. Several lists have been given out by the Department of Antiquities under the government Ministry of Natural Resources and Tourism(MNRT), Dar es Salaam Master Plan Consortium under the government Ministry of Lands, Housing and Human Settlement Development and by DARCH (Dar es Salaam Centre for Architectural Heritage) which is a private non-profit organization dealing with conservation of architectural heritage.

-The lists have been contradicting one another as they are not reconciled as one resulting in conflicts on managing and demolition of architectural heritage buildings in the Central Business District (CBD) of Dar es Salaam.

POINT 3

In map 1, there are several buildings named “Architectural Heritage Buildings”. Who named these buildings like this? Based on what?

Response 3

-List given out by the Department of Antiquities (DOA) in 1995 with 28 buildings to be under conservation whereby 27 of them are located at the CBD.

-List given out by the Department of Antiquities in 2006 with a total of 110 buildings which was controversial. A new list was supposed to be given out by the task force created but no list has been given out to the moment.

-List provided by DARCH (Dar es Salaam Centre for Architectural Heritage) with 58 buildings, 52 still standing and 6 buildings demolished.

-List by Dar es Salaam Master Plan Consortium 2016-2036 which has a total of 259 buildings.

-All the lists are considered valid something which brings contradiction in conserving architectural heritage buildings.

Based on: -The provision of lists has been guided by The Antiquities Act of 1964 and its amendment in 1979.

POINT 4

- Analyze as a cause, the probable lack of heritage protection measures in the master plan "Plan for remodelling the central area 2000", and not only as a consequence.

With this approach, it will be possible to better assess the value of the architectural heritage of the cited case study, and what are the specific problems that have not been preserved correctly. In short, it is considered necessary to provide a more detailed heritage evaluation to understand this duality between urbanization and architectural heritage, to be able to understand this proposal for future coexistence.

Response 4

-Lack of protective measures on the 2000 redevelopment plan on architectural heritage buildings as the focus of the redevelopment plan was to cater for the existing and future demands for infrastructure services, increasing daytime population, commercial and office space which was high and increasing, to have Corporate buildings of many stories as possible.

-The streets had become narrower and could not accommodate the populations.

POINT 5

Explain what the acronym CBD means, since it is not explained anywhere, and any international reader may not understand what it refers to. It could be explained in the introduction and in the rest of the paper, the acronyms could be used.

Response 5

The acronym has been defined in the Abstract and the Introduction fourth paragraph.

POINT 6

Explain why the date of 1967 is used as the beginning of the analysis (what milestone occurs on that date?) Since in the first periods, the transformation process does not materialize so strongly.

Response 6

-It was a period when changes in the built fabric of CBD started taking place after the independence of the country in 1961.

-The beginning of the establishment of policies to govern the newly independent nation.

-Threats of demolition of then-existing buildings had started occurring and the Antiquities Act of 1964 to protect heritage buildings were already in place. This includes indigenous buildings and those left by Arabs, German and British.

POINT 7

Be careful with proposing tourism as an exemplary measure to revitalize and preserve heritage, since mass tourism is being precisely one of the causes of the destruction of heritage in other parts of the world. It is recommended to specify sustainable and controlled tourism measures.

Response 7

Ecotourism which is sustainable and controlled tourism has been suggested instead of just “tourism”. That means the tourism should consider a balance between the amounts of activities which will not erode the conservation but it will activate the central business area with minimum interference with the existing.

POINT 8

The English language could be improved

In short, it is considered an interesting case study, and the dissemination of the research results may be useful for society and the scientific community, however, we consider it necessary to complement the urban and socio-economic analysis with a more detailed heritage approach.

Response 8

The entire paper has been sent to a professional proofreader for the English language. The new submission contains the proofread version

RESPONSE TO THE REVIEWERS COMMENTS-REVIEWER #2

POINT 1

Research is very valuable, but the text content is too superficial. There are only two visuals related to the research area and there are many historical and registered buildings that need to be evaluated. In the discussion section, a table should be created and the change in each of them should be determined by distinguishing between residential and commercial areas. In the discussion section, a table should be created and the change in each of them should be determined by distinguishing between residential and commercial areas. Especially the conclusion part is very superficial. The research is about the protection of historical sites, but in the conclusion part, conservation proposals are not developed in line with the determinations.

Response 1

Inserted in the revised Manuscript

Reviewer 2 Report

Research is very valuable but the text content is too superficial. There are only two visuals related to the research area and there are many historical and registered buildings that need to be evaluated. In the discussion section, a table should be created and the change in each of them should be determined by distinguishing between residential and commercial areas. In the discussion section, a table should be created and the change in each of them should be determined by distinguishing between residential and commercial areas. Especially the conclusion part is very superficial. The research is about the protection of historical sites, but in the conclusion part, conservation proposals are not developed in line with the determinations.

Author Response

RESPONSE TO THE REVIEWERS COMMENTS-REVIEWER #2

POINT 1

Research is very valuable, but the text content is too superficial. There are only two visuals related to the research area and there are many historical and registered buildings that need to be evaluated. In the discussion section, a table should be created and the change in each of them should be determined by distinguishing between residential and commercial areas. In the discussion section, a table should be created and the change in each of them should be determined by distinguishing between residential and commercial areas. Especially the conclusion part is very superficial. The research is about the protection of historical sites, but in the conclusion part, conservation proposals are not developed in line with the determinations.

Response 1

Inserted in the revised Manuscript

Round 2

Reviewer 1 Report

A more detailed description of the architectural heritage conservation management and its currently contradictions analysis has been explained in the paper revision. The explanation about the criteria for the selection of the architectural heritage buildings has been developed. However, It is not fully explained ifthere is a degree of protection for listed historic buildings that requires specific conservation measures or does not exist in the different organs mentioned. The comments are considered partially answered but with a enough level to be accepted.

The rest of the minor comments have been solved, and the English language has been improved.

Author Response

Comment

How it has been addressed

Comments from Reviewer #1

1.

 A more detailed description of the architectural heritage conservation management and its current contradictions analysis has been explained in the paper revision. The explanation about the criteria for the selection of the architectural heritage buildings has been developed. However, it is not fully explained if there is a degree of protection for listed historic buildings that require specific conservation measures or does not exist in the different organs mentioned. The comments are considered partially answered but with enough level to be accepted.

-Between the mentioned organs only the Dar es salaam Master Plan consortium has tried to establish the degree of conservation whereby among the 259 buildings listed it has documented the location and extent of alteration of 99 buildings among them but not yet under implementation.

 -The New Dar es Salaam Master plan 2012/2032 has given out a list of 259 buildings and mapped 92 buildings to be under restoration and renovation with their degrees i.e

Restoration type A;

-          Restoration of the façade

-          The philosophical re-building parts of the broken or demolished building

-          The conservation of the typological and functional schemes

-          The conservation of open spaces is directly related to the building

Restoration type B;

-          Restoration and reintroduction of the façade and demolition of the illegal and incoherent add ons

-          Maintenance of the original formal structure of the roof, floor and all other elements useful to maintaining or renewing the integrity of the morphology

-          Changing the functional plan and introduction of new technologies are allowed

Renovation

-          It includes interventions to transform the building to obtain a building, partly or entirely different from the original without exceeding the town planning standards.

-          Demolition and reconstruction of the entire structure; even if identical in terms of shape, volume and area, is considered as building renovation.

(FROM NEW DAR ES SALAAM MASTER PLAN)

Comments:

From this study demolition and alteration of architectural heritage buildings has been a problem due to;-

·         From 1967 to 2011 this degree had not yet been established hence there was no clarity.

·         After being established in 2012 the problem of miscoordination between responsible authorities (Ministry of Lands, Housing and Infrastructural Development and Ministry of Natural resources and Tourism) resulted in the continuation of demolition and restoration; hence, -

·         a participatory approach and coordination are suggested to have one system to coordinate the conservation of architectural heritage.

Comments from Reviewer #2

1.

In the conclusion section, recommendations for conservation should be developed

-Additional points and explanations on the “Policy implications” section have been added such as; -

·         Participatory approach policies involving owners and government to be formulated to protect few remaining buildings and cooperation from both sides in decisions and plans concerning land and building uses.

·          It was revealed in the study about the lack of a clear list of buildings under conservation, leading to contradiction on conservation issues. The responsible ministries, authorities, and stakeholders i.e (Department of Antiquities, DARCH (Dar es Salaam Centre for Architectural Heritage), Dar es Salaam Master Plan Consortium under Ministry of Lands, Housing and Human Settlement Development and Ministry of Natural Resources and Tourism) are advised to coordinate together and provide a clear list of buildings to avoid contradictions.

Reviewer 2 Report

In the conclusion section, recommendations for conservation should be developed.

Author Response

Thank you.
